# Enhancing Efficacy of a Brief Obesity and Eating Disorder Prevention Program: Long-Term Results from an Experimental Therapeutics Trial

**DOI:** 10.3390/nu15041008

**Published:** 2023-02-17

**Authors:** Eric Stice, Paul Rohde, Meghan L. Butryn, Christopher Desjardins, Heather Shaw

**Affiliations:** 1Department of Psychiatry and Behavioral Sciences, Stanford University, 401 Quarry Road Stanford, Stanford, CA 94305, USA; 2Oregon Research Institute, Springfield, OR 97477, USA; 3Department of Psychology, Drexel University, Philadelphia, PA 19104, USA; 4Department of Statistics, Saint Michaels College, Colchester, VT 05439, USA

**Keywords:** prevention, obesity, eating disorder, experimental therapeutics, response training

## Abstract

**Objective**: Test whether the efficacy of *Project Health*, an obesity/eating disorder prevention program, is improved by delivering it in single-sex groups and adding food response inhibition and attention training. **Method**: High-risk young adults (*N* = 261; *M* age = 19.3, 74% female) were randomized to (1) single-sex or (2) mixed-sex groups that completed food response inhibition and attention training or (3) single-sex or (4) mixed-sex groups that completed sham training with nonfood images in a 2 × 2 factorial design. **Results**: There was a significant sex-composition by training-type by time interaction; participants who completed single- or mixed-sex *Project Health* groups plus food response and attention training showed significant reductions in body fat over a 2-year follow-up, though this effect was more rapid and persistent in single-sex groups, whereas those who completed single- or mixed-sex *Project Health* groups plus sham training did not show body fat change. However, there were no differences in overweight/obesity onset over the follow-up. The manipulated factors did not affect eating disorder symptoms or eating disorder onset, but there was a significant reduction in symptoms across the conditions (within-condition *d* = −0.58), converging with prior evidence that *Project Health* produced larger reductions in symptoms (within-condition *d* = −0.48) than educational control participants. Average eating disorder onset over the 2-year follow-up (6.4%) was similar to that observed in *Project Health* in a past trial (4.5%). **Conclusions**: Given that *Project Health* significantly reduced future onset of overweight/obesity in a prior trial and the present trial found that body fat loss effects were significantly greater when implemented in single-sex groups and paired with food response and attention training, there might be value in broadly implementing this combined intervention.

## 1. Introduction

Obesity and overweight cause 3.4 million deaths annually, primarily due to cardiovascular diseases, cerebrovascular diseases, diabetes, and cancer [1], making broad implementation of prevention programs that reduce unhealthy weight gain a key public health priority. In 1998 the World Health Organization created the international standardized body mass index (BMI) definitions for overweight (BMI > 25 and <29.9) and obesity (BMI > 30) that were selected because those levels of adiposity significantly increased risk for weight-related morbidity and mortality [2]. It would be particularly useful to target young adults who ranged from 17–20 years in age because this developmental period is a particularly high-risk period for excess weight gain and excess weight gain that emerges during this period typically persists into adulthood [3]. Theoretically, this is a critical developmental period because for many young adults it is the first time they are completely responsible for their choices regarding the types and amounts of food that they consume and whether they engage in any physical activity. Three obesity prevention programs have significantly reduced future weight gain in young adults [4,5,6], but these interventions were extremely intensive, lasting from 8- to 28-months in duration, which would make broad implementation of these programs challenging and costly.

It would be ideal if such prevention programs also reduced eating disorder onset because eating pathology predicts weight gain [7,8] and mortality [9]. This is particularly important given the concerns that obesity prevention programs may increase risk for emergence of eating disorders [10].

Several prevention programs have been designed to reduce future onset of both obesity and eating disorders (e.g., [11,12]) but only one has prevented both public health problems. High risk adolescent female participants in the *Healthy Weight* prevention program showed a 53% reduction in obesity onset and a 60% reduction in eating disorder onset over a 3-year follow-up period and greater reductions in BMI and eating disorder symptoms through 3-year follow-up compared to controls and alternative interventions [13]. *Healthy Weight* participants were asked to make small incremental lifestyle changes to balance caloric intake with expenditure. The evidence that unhealthy weight control behaviors increase risk for future eating disorders [7,8] and that compensatory weight control behaviors are often the first behavioral symptom to emerge in eating disorders [14] suggests that helping at-risk youth to bring energy intake into balance with energy expenditure should reduce the eating pathology. In a second trial, *Healthy Weight* produced greater reductions in BMI and eating disorder symptoms in high-risk young women, prevented BMI gain through a 1-year follow-up for overweight participants, and reduced eating disorder onset by 60% over a 2-year follow-up compared to educational controls [15,16].

As prior research has shown a dissonance-based eating disorder prevention program successfully prevents eating disorder onset [13,17,18], in an effort to improve the efficacy of *Healthy Weight*, activities to produce dissonance about behaviors that cause unhealthy weight gain were added. The dissonance-based *Project Health* incorporates verbal, written, and behavioral exercises that evoke dissonance about unhealthy lifestyle behaviors; this theoretically increases the chances that participants match their attitudes with the stances they assumed in sessions, resulting in healthier lifestyle choices. *Project Health* produced smaller BMI increases and reduced overweight/obesity onset by 41% and 42% compared to *Healthy Weight* and educational video controls, respectively, through a 2-year follow-up for high-risk young adults; it also reduced eating disorder symptoms and marginally reduced (60%) future eating disorder onset over a 2-year follow-up, compared to controls [19].

In the present trial, two factors which were hypothesized to improve *Project Health’s* weight gain prevention effects were experimentally manipulated. First, because *Healthy Weight* produced weight gain prevention effects when implemented in female-only groups [13,15,16,20] but not when implemented in mixed-sex groups [19], we hypothesized that *Project Health* would produce larger body fat gain prevention effects when implemented with single- versus mixed-sex groups. Several interventions have shown greater efficacy when implemented in female-only versus mixed-sexed groups [21,22]. Researchers have theorized that this is because the greater commonality among participants in single-sex groups fosters greater group cohesion, trust, honesty, and willingness to share personal information. In *Project Health* sessions it was particularly important for participants to be honest about their lifestyle behaviors and whether they were successful in making the intended lifestyle changes. Additionally, conversations in mixed-sex dyads were interrupted more by both sexes than conversations in same-sex dyads [23], which could potentially reduce participation in dissonance-induction discussions. Please note that we focused on sex-at-birth rather than gender because the National Institutes of Health which funded this project requests a focus on sex in research they fund.

Adding food response inhibition and attention training to *Project Health* was also hypothesized to produce larger body fat gain prevention effects. Greater responsivity of brain reward valuation regions (striatum, orbitofrontal cortex) to high-calorie food images [24,25,26,27], greater attentional bias for high-calorie food [28], and lower inhibitory control in response to high-calorie foods, predicted future weight gain [29,30,31]. These data suggest that reducing reward and attention region responses to high-calorie foods and increasing inhibitory control via an intervention may decrease overeating. Go/no-go and stop-signal computer training, where participants were cued to repeatedly respond behaviorally to low-calorie food or non-food images, and to repeatedly inhibit behavioral responses to high-calorie food images, produced more weight loss compared to sham training [32,33]. Further, dot-probe tasks that train attention away from high-calorie foods and toward low-calorie foods reduced high-calorie food intake compared to sham training [34,35]. When overweight/obese adults completed a multifaceted response inhibition training with high-calorie foods and attend-away training from high-calorie foods, they had greater body fat loss, and showed a reduced fMRI-assessed reward region (putamen; mid insula) and attention region (inferior parietal lobe) response to, and palatability ratings and monetary valuation of, high-calorie foods versus sham training which used non-food images [36].

A 2-factorial design was used to test whether implementing single- versus mixed-sex *Project Health* groups and adding food response inhibition and attention training compared with sham training would enhance body fat gain prevention effects. The baseline comparison condition consisted of mixed-sex *Project Health* groups with sham response inhibition/attention training.

A report on the acute effects from this trial confirmed that *Project Health* produced significantly larger body fat loss by posttest when delivered in single- versus mixed-sex groups and when paired with the food response inhibition and attention training versus sham training [14]; and further, that these two factors interacted, producing the largest body fat loss for single-sex groups implemented with food response inhibition and attention training. Although the two manipulated factors did not affect eating disorder symptoms, there was a large reduction in symptoms across conditions (within-participant *d* = 0.78), that was larger than the within-condition symptom reduction (*d* = 0.54) observed in a past trial in which *Project Health* produced greater symptom reductions than educational controls [19]. The present report describes the intervention effects over a 2-year follow-up.

## 2. Methods

### 2.1. Participants and Procedures

At baseline, participants (*N* = 261) had a mean age of 19.3 years (*SD* = 0.8) and mean BMI of 24.5 kg/m^2^ (*SD* = 2.6). Three-quarters (74%) of the participants identified as female. The racial/ethnic composition of the sample was as follows: 74% White, 29% Asian, 7% Black, 2% Native Americans, 2% Pacific Islander, 10% Hispanic). Participants were recruited from October 2018 to March 2020 using mailings, flyers, and leaflets in two university communities. The intervention was advertised as teaching a healthy lifestyle to young adults with weight concerns. We focused on young adults with weight concerns based on evidence that such individuals are at risk for unhealthy weight gain [37]. See the acute-effects report [14] for details regarding screening, consent, and inclusion/exclusion criteria. Figure 1 describes participant flow. Participants completed assessments at pretest, posttest, and 6-, 12-, and 24-month follow-ups. Participants were randomized to condition after completion of the pretest assessment. Research staff members who completed assessments were blind to participants’ assigned conditions. 

### 2.2. Interventions

*Project Health*. The *Project Health* intervention was delivered in a closed group format, with 6 to 10 participants assigned to each group. Groups met weekly for 6 weeks, with each group session lasting 60 min. Each session was facilitated by a graduate student in psychology or a related field.

The goal of the intervention was to help participants make changes to dietary intake and exercise, to bring energy intake into balance with energy expenditure. The facilitators did not prescribe specific behavioral goals on a weekly basis, rather they encouraged participant autonomy in selecting the specific ways in which they would like to change their eating and activity behaviors each week. To promote cognitive dissonance, activities were completed in session and assigned for homework in which participants identified and reflected on the costs of overeating, sedentary behaviors, and obesity. Activities also encouraged participants to reflect on the benefits of choosing healthy foods, engaging in physical activity, and maintaining a healthy body weight.

Participants who were randomly assigned to the “single-sex group” condition attended groups that consisted only of participants of the same sex (based on reported sex at birth). In single-sex groups, facilitators were the same sex as the participants. In the “mixed-sex group” condition, each group included at least two participants of each sex. In mixed-sex groups, the facilitator could be of either sex. Individuals who reported non-cisgender identity were assigned to mixed-sex groups.

Facilitators attended a 6-h training session to promote intervention competence and fidelity to the intervention protocol. All group sessions were videotaped. Supervisors watched each of the first six sessions that each facilitator conducted, rated competence and fidelity using validated scales, and provided written feedback to the facilitator that was based on the ratings [19]. Supervisors continued to watch and make ratings for 50% of a facilitator’s subsequent sessions, providing feedback based on these ratings.

*Food Response Inhibition and Attention Training*. Each week, all participants were asked to remain on site after the *Project Health* session ended, to complete 25 min of additional intervention tasks on a computer. Participants first completed a brief dissonance-based motivational enhancement task, where they wrote about topics, such as the importance of achieving and maintaining a healthy body weight. In addition, they completed four computer-based training tasks that each lasted 5 min: stop-signal training, go/no-go training, dot probe training and visual search training. In the stop-signal and go/no-go tasks, participants were cued to repeatedly respond behaviorally with a button press to low-calorie foods and repeatedly inhibit a response to high-calorie foods. The dot-probe task showed a probe near low-calorie foods on 90% of trials, thereby reinforcing looking at low-calorie foods and training visual attention away from high-calorie foods. The visual search task showed participants an array of images that included one low-calorie food (a fruit or vegetable) within an array of high-calorie foods. The task trains participants to ignore the images of high-calorie foods and attend to the low-calorie foods.

The high-calorie food images used in the trainings were customized to each participant. Participants were asked about the types of foods they often overate. The three categories of high-calorie foods that they overate most frequently were identified. Participants then rated images of foods within these three categories for palatability. The 40 high-calorie foods rated highest for palatability were the ones shown to each participant in the trainings. 

The sham response inhibition training intervention was identical to the food response training, but used images of birds and flowers, rather than foods [36]. To promote credibility, participants in both conditions were told that these tasks were designed to improve response inhibition, and thereby reduce impulsive overeating.

### 2.3. Measures

*Body fat*. We used air displacement plethysmography (ADP) via the Bod Pod to assess percent body fat. ADP percent body fat shows high test-retest reliability (*r* = 0.92–0.99) and correlates with DEXA and hydrostatic weighing estimates (*r* = 0.98–0.99) [38]. Participants were asked to not consume any food or beverages (other than water) or nicotine products for 3+ h, and refrain from vigorous exercise for 24+ h prior to each measurement.

*Eating disorder symptoms.* The Eating Disorder Diagnostic Interview [EDDI; 19] assessed DSM-5 eating disorder symptoms and diagnoses. Items assessing symptoms in the past month were summed to form a composite at each assessment. This composite has shown internal consistency (α = 0.92), inter-rater agreement (*r* = 0.93), 1-week test-retest reliability (r = 0.95), and predictive validity [18]. EDDI diagnoses have shown 1-week test-retest reliability (κ = 0.79), inter-rater agreement (κ = 0.75), and predictive validity [18].

### 2.4. Data Analyses

To examine continuous outcome change by single-sex versus mixed-sex group and food response and attention training versus sham training, we fitted generalized additive models (GAM). GAMs were used because outcome change was nonlinear. These analyses were intent-to-treat and included all participants randomized to the conditions. Because participants in the four conditions differed on baseline percent body fat (F_3, 251_ = 5.63, *p* = 0.001), all analyses were controlled for baseline percent body fat. Interactions for GAMs were examined by fitting separate smooth functions of time for each intervention condition. To examine interactions of time with the four conditions, we fitted five models. First, we fitted a model with a three-way interaction between time, group-type, and response training-type (all models included lower order interactions and main effects) and random effects for intercept and slope. Second, we fitted a model with a two-way interaction between time and group and time and response type modeled as random effects. Third, we fitted each two-way interaction separately in their own model. Finally, we fitted a model with no interactions with time. These models were compared using the Akaike Information Criterion (AIC) [39]. Models with the lowest AIC best fitted the data and models that differ by less than seven were considered to have a comparable fit [40], in which case, we selected the more parsimonious model. We also conducted complier analyses that only included participants who attended five or six of the *Project Health* sessions, to provide a description of the intervention effects for participants who completed most of the intervention.

To test whether the onset of overweight/obesity or eating disorders differed by condition, Cox proportional hazard models were fitted. Participants were classified as underweight, healthy, overweight, or obese used the criteria in Gallagher et al. [41]. Threshold and subthreshold eating disorders are operationalized in Stice et al. [18]. Participants who were obese at baseline were excluded from the model testing for differences in new onset of overweight or obesity across conditions. Participants who had a subthreshold eating disorder at baseline were excluded from the model testing for differences in onset of threshold or subthreshold eating disorders across conditions.

## 3. Results

Participants with versus without complete data did not differ on any of the outcomes at baseline. Table 1 shows summary statistics of outcomes by condition. Analyses confirmed that participants assigned to the four conditions did not differ significantly on any study variable, including body fat. Table 2 shows the results of the AIC model comparisons.

For percent body fat, the model with the three-way interaction between group-type, response training, and time, fitted better than simpler models. Results for the percent body fat GAM are shown in Table 3 and Figure 2. Only single-sex groups plus food response/attention training (*p* = 0.010) and mixed-sex groups plus food response/attention training (*p* = 0.010) had statistically significant reductions in body fat; change in body fat did not differ from zero over follow-up in the single-sex and mixed-sex groups plus sham training. Figure 2 shows the GAM estimated change in percent body fat during follow-up along with 95% confidence intervals. *Project Health* produced significantly greater reductions in body fat when paired with food response/attention training, with single-sex groups showing more rapid body and persistent fat loss than mixed-sex groups. There was no evidence suggesting that body fat change by condition significantly differed for males versus females (all *p*-values > 0.05).

Complier analysis revealed that, averaging across conditions, participants who attended five or more *Project Health* sessions showed greater body fat loss by the 2-year follow-up than participants who attended fewer sessions (*t* = 2.23, df = 253, *p* = 0.032, partial η^2^ = 0.02). Participants with a high compliance lost 1.44% more body fat. For the eating disorder symptom composite, the best fitting model contained a two-way interaction of time and response training; however, this fit was not substantially better than the model containing no interaction with time.

Onset of overweight/obesity and of any eating disorder over follow-up did not differ across conditions (Table 4).

## 4. Discussion

This study was designed to test if the efficacy of *Project Health* could be enhanced by conducting groups in a single-sex, rather than mixed-sex, format, and supplementing group sessions with food response inhibition and attention training. We found a significant three-way interaction between group composition, response training, and time, in which participants who completed *Project Health* in either single- or mixed-sex groups paired with food response inhibition/attention training showed significant body fat loss effects over a 2-year follow-up, but not participants who completed *Project Health* groups paired with sham training. The body fat loss effect occurred earlier for single-sex groups and was more persistent than for mixed-sex groups. However, there were no significant differences in onset of overweight/obesity across condition. The body fat loss effects over the 2-year follow-up converge with body fat loss effects by posttest from this trial, reported previously [14], and with a trial that evaluated food response and attention training in isolation [36]. Importantly, this is the first trial to provide evidence that food response inhibition/attention training produced body fat loss effects over a long-term follow-up (2-years) relative to sham training. The fact that participants only spent 2 h completing this training makes these findings particularly noteworthy. Because this trial did not have a minimal intervention control condition, it is important to note that young adults with weight concerns who were randomized to an educational video control in our last trial showed increases in body weight over the 2-year follow-up [19], implying that *Project Health* reduced the weight gain trajectory shown by these at-risk youths even when paired with sham training. Moreover, consistent with the acute effects report [14], complier analyses indicated that participants who attended five of more *Project Health* sessions showed significantly more body fat loss, suggesting that improving session attendance should improve efficacy. Given that *Project Health* has produced larger reductions in body weight than a minimal control and an alternative obesity/eating disorder prevention program [19], results imply that pairing *Project Health* with food response inhibition and attention training might produce the strongest weight gain prevention effects.

Averaging across the four *Project Health* conditions, there was a significant within-condition reduction in eating disorder symptoms (*p* < 0.001, *d* = −0.58). This effect was similar to the within-condition reduction in symptoms for *Project Health* (*d* = −0.48) in our last trial, which was significantly larger than the within-condition eating disorder symptom reduction for video controls [19]. Averaging across conditions, 6.4% of participants showed eating disorder onset over the 2-year follow-up, which was similar to the 4.5% onset over the 2-year follow-up for *Project Health* and lower than the 10% onset observed previously for educational controls [19]. The evidence that participants who completed *Project Health* showed reductions in eating disorder symptoms and lower onset of eating disorders than observed in control conditions in past trials is important given concerns that obesity prevention programs might increase the risk for eating disorders [10]. However, changes in eating disorder symptoms and eating disorder onset over the follow-up did not significantly differ in participants who were assigned to single-sex versus mixed-sex group. The discussions and activities in *Project Health* groups were designed to promote cognitive dissonance about lifestyle behaviors that increase risk for excess weight gain (e.g., consumption of high-calorie foods and sedentary behavior). Research has suggested that activities that create dissonance about pursuing the thin appearance ideal, a potent risk factor for eating disorder, are more effective in preventing eating disorders (13, 17, 18). In this way, it is not surprising that the effect of the intervention on eating disorder outcomes did not depend on sex composition of one’s group. Similarly, changes in eating disorder symptoms and rate of eating disorder onset did not significantly differ in those who did versus those who did not receive food response inhibition/attention training, perhaps because the intervention targets of this element of the intervention (elevated reward region response to high-calorie foods, attentional bias for high-calorie foods, and deficits in inhibitory control) have not been shown to increase risk for future onset of eating disorders. In sum, this study confirmed that the *Project Health* intervention can reduce eating disorder symptoms and reduce risk of eating disorder onset, but found no evidence that these effects are further enhanced by conducting single-sex groups or adding food response inhibition/attention training to the intervention package.

Regarding limitations, the retention rate in this study was lower than it was in other evaluations of *Project Health,* largely because the COVID-19 pandemic prevented us from conducting in-person follow-up assessments for many participants. In addition, males were under-represented in the study sample. Further, the trial was designed such that all participants received the *Project Health* intervention, because the aim of the study was to experimentally test the effects of group composition and response inhibition/attention training. There was no condition in which participants received a usual care or placebo intervention, and thus it is not possible to draw conclusions about how much benefit the optimized intervention package might have, relative to some type of minimal intervention control condition.

The latter limitation of the study design points to an important future direction for this work, which is to conduct a clinical trial in which young adults at risk of weight gain are randomly assigned to either (1) the optimized version of the *Project Health* intervention (i.e., conducted in single-sex groups and with food response inhibition/attention training) or (2) a comparison condition such as an obesity education intervention. With such a design, future research will be able calculate the intervention benefit in a way that can allow straightforward comparisons with other prevention programs. At this stage of research, it is reasonable to begin studying approaches for supporting dissemination of this weight gain/eating disorder prevention program for high-risk young adults. Future research should also test methods of enhancing session attendance, given evidence that the intervention effects on body fat were greatest among participants who received the greatest dose of the intervention. In addition, it would be useful to evaluate the effect of this prevention program for other populations at risk of excess weight gain such as adults with prediabetes, overweight pregnant women, and adults from racial and ethnic groups with an elevated prevalence of overweight and obesity (e.g., Black, Hispanic, and Native American adults). It might also be useful to evaluate this obesity/eating disorder prevention program among adolescents at high-risk of obesity, such as those with a parental history of being overweight or obesity.

Although several obesity prevention programs have significantly reduced future weight gain in young adults in randomized trials [5,6], these interventions were intensive, lasting from 8- to 28-months. Given that *Project Health* is briefer, has significantly reduced future onset of both overweight/obesity and eating disorders [19], and produced significantly stronger body fat gain prevention effects when paired with food response inhibition and attention training, it might be valuable to broadly implement this dual obesity and eating disorder prevention program.

## Figures and Tables

**Figure 1 nutrients-15-01008-f001:**
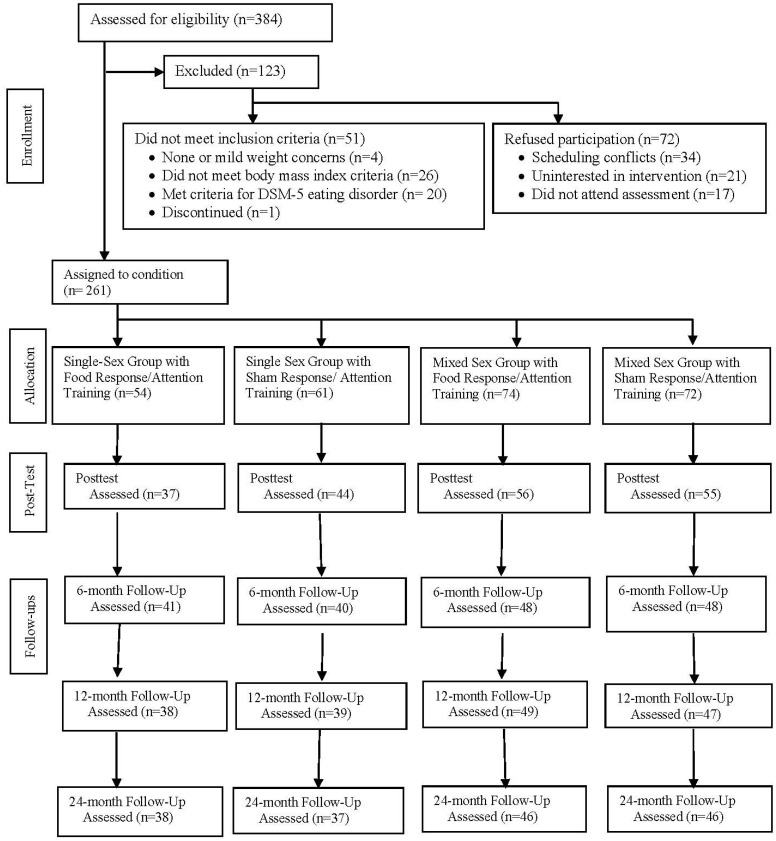
Participant flow through the study.

**Figure 2 nutrients-15-01008-f002:**
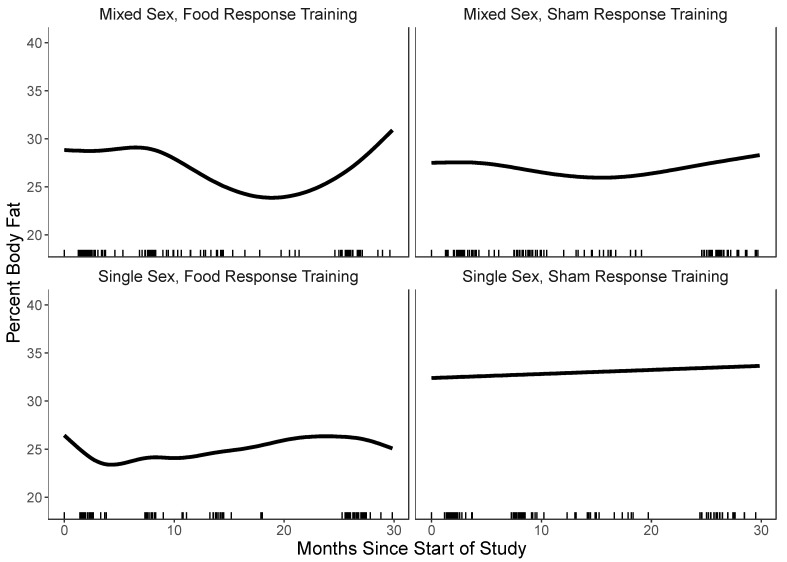
All panels present change in percent body fat over the 2-year follow-up in each of the four conditions. Model Predicted Change in Percent Body Fat by Condition.

**Table 1 nutrients-15-01008-t001:** Descriptive Summary for Study Variables by Condition.

				Follow-Up		
Condition	Outcome	Pretest	Posttest	6-Month Follow-Up	1-Year Follow-Up	2-Year Follow-Up
Single-sex group with food response training	Body Fat	26.84 (7.33)	22.93 (6.62)	23.23 (8.15)	26.95 (6.58)	24.13 (8.76)
	Eating Disorder Symptoms	12.46 (11.37)	5.73 (5.92)	5.12 (5.3)	5.43 (5.3)	6.11 (6.98)
Single-sex group with sham training	Body Fat	32.83 (6.74)	32.71 (6.57)	33.98 (7.36)	31.79 (6.11)	32.9 (9.28)
	Eating Disorder Symptoms	11.92 (8.82)	6.91 (6.28)	5.82 (4.53)	4.44 (4.25)	6.68 (10.59)
Mixed-sex group with food response training	Body Fat	29.2 (9.55)	29.61 (7.65)	29.94 (9.4)	23.53 (11.16)	25.73 (9.67)
	Eating Disorder Symptoms	12.56 (11.6)	6.88 (6.06)	6.45 (5.92)	7.17 (7.71)	5.52 (5.42)
Mixed-sex group with sham training	Body Fat	27.72 (9.84)	29.21 (9.13)	27.62 (75)	28.59 (8.12)	27.83 (9.23)
	Eating Disorder Symptoms	11.92 (11.89)	7.87 (13.17)	6.07 (5.75)	3.98 (3.13)	5.53 (5.53)

**Table 2 nutrients-15-01008-t002:** Akaike Information Criteria (AIC) for Candidate Models by Outcome.

Percent Body Fat					
	T:G:RT	T:G and T:RT	T:G	T:RT	No Interactions
df	**363.80**	362.57	358.88	357.04	357.63
AIC	**3533.96**	3543.74	3555.18	3572.82	3573.49
ΔAIC	**0**	9.78	21.22	38.86	39.53
**Eating Disorder Symptoms**					
	T:G:RT	T:G and T:RT	T:G	T:RT	No Interactions
df	264.02	262.80	262.36	261.60	**261.04**
AIC	4917.85	4917.25	4917.95	4916.30	**4916.72**
ΔAIC	1.55	0.95	1.65	0	**0.42**

Note. T:G:RT refers to the three-way interaction of time, group, and response training; T:G to the interaction of time and group; and T:RT to the interaction of time and response training. ΔAIC refers to the difference in the current model’s AIC and the model with the smallest AIC. The bold-faced model refers to the selected model based on AIC.

**Table 3 nutrients-15-01008-t003:** Coefficient Estimates for the Body Fat Model.

Fixed Effects	Estimate	SE	t-Statistic	*p*-Value
Intercept	28.18	1.08	26.00	<0.001
Single-Sex	−2.74	1.66	−1.65	0.100
Sham training	−0.87	1.55	−0.56	0.573
Single Sex:sham training	8.16	2.32	3.51	0.001
Smooth Functions		EDF	F-statistic	*p*-value
Time		0.80	0.80	0.810
Time:Mixed-Sex, Food Response Training		4.07	3.79	0.010
Time:Single-Sex, Food Response Training		4.60	2.91	0.010
Time:Mixed-Sex, Sham Training		2.71	0.09	0.208
Time:Single-Sex, Sham Training		0.80	0.80	0.790
Random Effects		EDF	F-statistic	*p*-value
Random Intercept		235.63	83.94	<0.001
Random Slope		110.28	52.376	<0.001

**Table 4 nutrients-15-01008-t004:** Onset of any Eating Disorder or Overweight/Obesity during the Study by Condition.

Condition	Did Not Develop Eating Disorder	Developed Eating Disorder
Single-sex group with food response training	45	6
Single-sex group with sham training	55	3
Mixed-sex group with food response training	62	3
Mixed-sex group with sham training	53	3
Condition	Did Not Transition to Unhealthy Weight	Transitioned to Unhealthy Weight
Single-sex group with food response training	30	8
Single-sex group with sham training	37	8
Mixed-sex group with food response training	38	9
Mixed-sex group with sham training	43	10

## Data Availability

Data can be requested from the first author.

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
