# Peer review of "Enhancing Efficacy of a Brief Obesity and Eating Disorder Prevention Program: Long-Term Results from an Experimental Therapeutics Trial"

_nutrients, 2023, doi:10.3390/nu15041008_

Round 1
Reviewer 1 Report
The authors are to be commended for addressing a very important area in the prevention of EDs and obesity. The paper is, in general, clearly written and the findings are promising for the field.
Intro
- This section would benefit from some broader discussion around latest obesity definitions (e.g., from WHO) not being so focused on weight/BMI but rather negative health/quality of life impacts. There should also be some discussion of weight stigma and the links to eating disorders. Please consider bringing up these topics again in the Discussion when thinking about the future of the field of ED/Obesity.
- Line 70: The statement that single-sex groups promote cohesion, trust, honestly and willingness needs some further discussion. It would be helpful to draw on the literature or theory that helped establish the relevance to the study at hand. It would helpful for the reader to know if there are differences between sex and gender on this topic considering the study assigned groups based on reported sex at birth.
Methods
- It would be helpful for the reader to include more information in the manuscript beyond "See the acute-effects report [8] for details regarding screening, consent, and inclusion/exclusion criteria".
- A rationale for the use of young university adults would be beneficial. If this is a result of convenience sampling, then a statement should be included in the limitations of the study, including implications of potentially higher socioeconomic groups on the intervention. If not, a discussion in the introduction supporting why this specific age range is important to target for early intervention would be helpful to the reader.
Discussion
- Further explanation to your comment (line 257) that it seems logical that the dissonance-induction activities would have only impacted body fat is needed.
- There is a good discussion around how the present study fits in with literature to date, however what is missing is a discussion around the potential clinical implications and implementation of the intervention. What would implementation of this intervention look like considering the present study was conducted in young university students with a mean age of 19 years? Could future research assess the efficacy of this intervention in older populations?
- The final paragraph of the discussion (line 286) comments how previous obesity intervention programs are intensive in length, a limitation which the present program overcomes, however this is not mentioned so much in other sections of the manuscript. It would be helpful to see this point integrated in other sections and perhaps how this will impact the efficacy of the implementation of the program.
Author Response
Dear Professor Stice,
Thank you again for your manuscript submission:
Manuscript ID: nutrients-2174684
Type of manuscript: Article
Title: Enhancing Efficacy of a Brief Obesity and Eating Disorder Prevention
Program: Long-Term Results from an Experimental Therapeutics Trial
Authors: Eric Stice *, Paul Rhode, Meghan L Butryn, Christopher D.
Desjardins, Heather Elizabeth Shaw
Received: 5 January 2023
E-mails: estice@stanford.edu, paulr@ori.org, mlb34@drexel.edu,
cddesjardins@gmail.com, hshaw2@stanford.edu
Submitted to section: Nutrition in Women,
https://www.mdpi.com/journal/nutrients/sections/Nutrition_Women
Eating Disorders, and Nutritional Beliefs, Trends or Practices
https://www.mdpi.com/journal/nutrients/special_issues/eating_disorder_nutrition
Your manuscript has now been reviewed by experts in the field. Please find
your manuscript with the referee reports at this link:
https://susy.mdpi.com/user/manuscripts/resubmit/a765d95a486f9f607ac33d958bf98107
Please reduce the similarity according to the report attached and also add
more content, the length of the main body (from introduction to conclusion
section) should be above 4000 words. As requested, we have edited the manuscript to reduce redundancy with the acute-effects report from this trial.
Please revise the manuscript according to the referees' comments and upload
the revised file within 10 days.
Please use the version of your manuscript found at the above link for your
revisions.
(I) Please check that all references are relevant to the contents of the
manuscript.
(II) Any revisions to the manuscript should be marked up using the “Track
Changes” function if you are using MS Word/LaTeX, such that any changes can
be easily viewed by the editors and reviewers.
(III) Please provide a cover letter to explain, point by point, the details
of the revisions to the manuscript and your responses to the referees’
comments.
(IV) If you found it impossible to address certain comments in the review
reports, please include an explanation in your appeal.
(V) The revised version will be sent to the editors and reviewers.
Do not hesitate to contact us if you have any questions regarding the
revision of your manuscript. We look forward to hearing from you soon.
Kind regards,
Thank you for the opportunity to revise our manuscript to address the comments from the reviewers. We have detailed below how we responded to comments from the reviewers. We tried to use track-changes, but the alternations we made to reduce redundancy were so extensive that it was difficult to read. Accordingly, we reflected edits to the manuscript in blue font, which is a little easier to read. – Eric Stice
R1
The authors are to be commended for addressing a very important area in the prevention of EDs and obesity. The paper is, in general, clearly written and the findings are promising for the field.
Intro
- This section would benefit from some broader discussion around latest obesity definitions (e.g., from WHO) not being so focused on weight/BMI but rather negative health/quality of life impacts. There should also be some discussion of weight stigma and the links to eating disorders. Please consider bringing up these topics again in the Discussion when thinking about the future of the field of ED/Obesity. In response, we now note that the BMI cut-points for overweight and obesity were originally selected because they were associated with significant elevations in morbidity and mortality (p. 3). We agree it is important to note this because many people do not seem to be aware of this history. In addition, we note that people have posited that weight stigma might increase risk for onset of eating disorders and discuss evidence for this assertion (p. 3), as requested. Further, we circle back to these topics in the discussion section (pp. 13-14).
- Line 70: The statement that single-sex groups promote cohesion, trust, honestly and willingness needs some further discussion. It would be helpful to draw on the literature or theory that helped establish the relevance to the study at hand. It would helpful for the reader to know if there are differences between sex and gender on this topic considering the study assigned groups based on reported sex at birth. As requested, we expanded our discussion of the evidence that led us to hypothesize that Project Health would show greater weight gain prevention/weight loss when implemented in single-sex versus mixed-sex groups (p. 5). In addition, we now note that we focused on sex-at-birth rather than gender because the National Institutes of Health which funded this study requests that researchers focus on sex in funded research studies (p. 5).
Methods
- It would be helpful for the reader to include more information in the manuscript beyond "See the acute-effects report [8] for details regarding screening, consent, and inclusion/exclusion criteria". We did not repeat these details because the editor requested that we minimize redundancy between the text in this manuscript and previously published papers.
- A rationale for the use of young university adults would be beneficial. If this is a result of convenience sampling, then a statement should be included in the limitations of the study, including implications of potentially higher socioeconomic groups on the intervention. If not, a discussion in the introduction supporting why this specific age range is important to target for early intervention would be helpful to the reader. In response, we have clarified that we intentionally targeted young adults who were between 17 and 20 years of age because this is a particularly high-risk developmental period for weight gain because for many young adults this is the first time that they are entirely responsible for making all of the decisions regarding the types of amounts of foods consumed and whether they engage in physical activity (p. 3). Indeed, this was one of the primary reasons that the National Institute of Child Health and Human Development (NICHD) funded this project rather than the National Institute of Diabetes and Digestive and Kidney Diseases (NIDDK).
Discussion
- Further explanation to your comment (line 257) that it seems logical that the dissonance-induction activities would have only impacted body fat is needed. As requested, we have expanded our discussion of why it was logical that the activities in the sessions, which were designed to induce dissonance regarding lifestyle behaviors that contribute to excessive weight gain would not be expected to reduce eating disorder symptoms, noting that it would be necessary to reduce risk factors that predict future onset of eating disorders to reduce eating disorder symptoms and eating disorder onset (pp. 13-14).
- There is a good discussion around how the present study fits in with literature to date, however what is missing is a discussion around the potential clinical implications and implementation of the intervention. What would implementation of this intervention look like considering the present study was conducted in young university students with a mean age of 19 years? Could future research assess the efficacy of this intervention in older populations? As suggested, we now say that a useful direction for future research will be to evaluate the effects of this obesity/eating disorder prevention program for other adult populations at high-risk for excess weight gain, such as adults with prediabetes, pregnant women, and individuals from racial and ethnic groups with elevated prevalence of overweight and obesity (pp. 15-16).
- The final paragraph of the discussion (line 286) comments how previous obesity intervention programs are intensive in length, a limitation which the present program overcomes, however this is not mentioned so much in other sections of the manuscript. It would be helpful to see this point integrated in other sections and perhaps how this will impact the efficacy of the implementation of the program. We had omitted that line of the argument because of the word limits, but agree that is would be important to note this topic in the introduction, which we now do (p. 3).
Reviewer 2 Report
This manuscript examined the efficacy of Project Health when (1) implemented in single-sex groups and (2) food response inhibition and attention training is added to the protocol. The authors found that participants who completed Project Health in either single or mixed sex groups paid with food response inhibition/attention training showed significant body fat loss effects over 2-year follow up, but not those participants who received Project Health with sham training. The authors did not find any change in onset of overweight/obesity or eating disorder symptoms according to intervention condition.
Overall, the manuscript is clear and there is value in reporting that this intervention helps with moderate fat loss, if that is what participants desire and what is practically and clinically significant. However, I am not convinced, based on this manuscript's framing of the results, that fat loss is equivalent to improved health, quality of life, disordered eating, etc. I think the manuscript would benefit from more discussion of the implications of the findings for individual health. I describe this concern, and others, below.
Abstract
· “Implementing Project 26 Health in single-sex groups with food response and attention training reduced body fat loss as well 27 as eating disorder symptoms suggesting that there might be value in broadly implementing this 28 combined intervention.” – This seems misleading to say, given how obesity/overweight onset over follow-up did not differ between the groups.
Introduction
“To improve the efficacy of Healthy Weight we added activities that create cognitive 53 dissonance about behaviors that cause unhealthy weight gain.” – This sentence felt out of place. The authors go from introducing the background on Healthy Weight to then talking in first person about modifications they made to it. Can the authors add a heading before this sentence that helps transition to introducing the present study? Or re-phrase to keep the introduction consistently in first person (or not).
“First, the fact that Healthy Weight intervention produced weight gain prevention effects when implemented in female-only groups [7, 9 10, 14] but not in mixed-sex groups [13]” – How does the present study differ from previous research? I thought one of the aims of this study (according to the abstract) is to examine single- versus mixed-sex groups? Scrolling down in the intro, it appears that actually this report is just an update on a previous manuscript, and reports on findings at 2-year follow up. This needs to be stated earlier in the manuscript and also in the abstract. It is important to note that this is not the first report of the initial findings and that the main point of this manuscript is the follow-up piece.
Last paragraph of intro: The authors should make it clearer that by “generic” they mean “sham” unless I am misunderstanding something.
Methods
Were those who identified outside of the gender binary excluded? What about transgender participants?
Results/Discussion
“Participants were excluded from these 186 analyses if they were underweight or obese at baseline or had an eating disorder at base- 187 line.” -weren’t prospective participants with eating disorders not permitted to sign up for the study?
So the single-sex group with food response training had double the amount of eating disorder diagnoses at follow up than the other groups – what do the authors make of this? To me, it sounds a lot like training people to avoid high calorie foods and training with other same-sex people who value shape/weight might predict disordered eating. I would appreciate a compelling counter-argument.
Could it be that training people to prefer lower-calorie foods only works to reduce weight or prevent weight gain in the short term because it is essentially selling them a diet, which have been shown to work in the short-term? Do the authors have any other ideas to explain why the enhanced intervention did not prevent crossover into the overweight or obesity category? It is important to comment on this.
Why should we care about fat loss if the crossover to obesity was not prevented in the Project Health with food response inhibition/attention training relative to the other groups? Is a small amount of fat loss notable if it does not actually prevent the thing we are supposedly worried about (obesity)? I would find such results more compelling if they were accompanied by some other metric of wellbeing relative to weight – like, increased body satisfaction, elevated self-esteem, etc. The way this discussion reads, it’s as if we are prioritizing a few pounds lost over the purported practical/clinical value of not having obesity. I would appreciate if the authors would re-frame their stance to reflect this reality.
Author Response
R2
This manuscript examined the efficacy of Project Health when (1) implemented in single-sex groups and (2) food response inhibition and attention training is added to the protocol. The authors found that participants who completed Project Health in either single or mixed sex groups paid with food response inhibition/attention training showed significant body fat loss effects over 2-year follow up, but not those participants who received Project Health with sham training. The authors did not find any change in onset of overweight/obesity or eating disorder symptoms according to intervention condition.
Overall, the manuscript is clear and there is value in reporting that this intervention helps with moderate fat loss, if that is what participants desire and what is practically and clinically significant. However, I am not convinced, based on this manuscript's framing of the results, that fat loss is equivalent to improved health, quality of life, disordered eating, etc. I think the manuscript would benefit from more discussion of the implications of the findings for individual health. I describe this concern, and others, below.
Abstract
- “Implementing Project 26 Health in single-sex groups with food response and attention training reduced body fat loss as well 27 as eating disorder symptoms suggesting that there might be value in broadly implementing this 28 combined intervention.” – This seems misleading to say, given how obesity/overweight onset over follow-up did not differ between the groups. In response, we clarified the concluding statement to read: Given that Project Health significantly reduced future onset of overweight/obesity in a prior trial and the present trial found that the body fat loss effects were significantly greater when Project Health is implemented in single-sex groups and paired with food response and attention training, there might be value in broadly implementing this combined intervention (p. 2).
Introduction
“To improve the efficacy of Healthy Weight we added activities that create cognitive 53 dissonance about behaviors that cause unhealthy weight gain.” – This sentence felt out of place. The authors go from introducing the background on Healthy Weight to then talking in first person about modifications they made to it. Can the authors add a heading before this sentence that helps transition to introducing the present study? Or re-phrase to keep the introduction consistently in first person (or not). In response, we have edited the introduction to improve the logical flow and no longer write in the first person in the identified section (p. 4).
“First, the fact that Healthy Weight intervention produced weight gain prevention effects when implemented in female-only groups [7, 9 10, 14] but not in mixed-sex groups [13]” – How does the present study differ from previous research? I thought one of the aims of this study (according to the abstract) is to examine single- versus mixed-sex groups? Scrolling down in the intro, it appears that actually this report is just an update on a previous manuscript, and reports on findings at 2-year follow up. This needs to be stated earlier in the manuscript and also in the abstract. It is important to note that this is not the first report of the initial findings and that the main point of this manuscript is the follow-up piece. In response, we now make it clear that we described the acute effects of this trial in an earlier report and that the purpose of the present manuscript was to describe the long-term effects (pp. 6-7).
Last paragraph of intro: The authors should make it clearer that by “generic” they mean “sham” unless I am misunderstanding something. To improve clarity, we decided to just use the term sham training and clarify what we meant by this term when we first used it (p. 2).
Methods
Were those who identified outside of the gender binary excluded? What about transgender participants? We had stated that: “Individuals who reported non-cisgender identity were assigned to mixed-sex groups.” (p. 8). In the acute effects report we had noted that there were 2 nonbinary participants and 4 transgender participants in this sample.
Results/Discussion
“Participants were excluded from these 186 analyses if they were underweight or obese at baseline or had an eating disorder at base- 187 line.” -weren’t prospective participants with eating disorders not permitted to sign up for the study? The acute effects report noted that we excluded participants with current anorexia nervosa, bulimia nervosa, or binge eating disorder but that there were 12 cases of subthreshold bulimia nervosa and 4 cases of subthreshold binge eating disorder. In response, we have clarified that we excluded participants with subthreshold eating disorders at baseline in the model testing for differences in new onset of threshold or subthreshold eating disorders (p. 11).
So the single-sex group with food response training had double the amount of eating disorder diagnoses at follow up than the other groups – what do the authors make of this? To me, it sounds a lot like training people to avoid high calorie foods and training with other same-sex people who value shape/weight might predict disordered eating. I would appreciate a compelling counter-argument. We had reported that onset of eating disorders in each of the conditions did not differ significantly. Per convention, this means that the variation in eating disorder onset was not reliable and most likely due to chance variation.
Could it be that training people to prefer lower-calorie foods only works to reduce weight or prevent weight gain in the short term because it is essentially selling them a diet, which have been shown to work in the short-term? Do the authors have any other ideas to explain why the enhanced intervention did not prevent crossover into the overweight or obesity category? It is important to comment on this. The results from the current randomized trial found that 2-hours of food response and attention training produced reductions in body fat that persisted through 2-year follow-up. That is, food response and attention training did not just reduce weight gain in the short-term. We should also note that Project Health expressly discourages transitory dieting and instead promotes lasting lifestyle reductions in intake of high-calorie foods.
Why should we care about fat loss if the crossover to obesity was not prevented in the Project Health with food response inhibition/attention training relative to the other groups? Is a small amount of fat loss notable if it does not actually prevent the thing we are supposedly worried about (obesity)? I would find such results more compelling if they were accompanied by some other metric of wellbeing relative to weight – like, increased body satisfaction, elevated self-esteem, etc. The way this discussion reads, it’s as if we are prioritizing a few pounds lost over the purported practical/clinical value of not having obesity. I would appreciate if the authors would re-frame their stance to reflect this reality. We had stated in the introduction that a previous trial found that Project Health produced a 41% and 42% reduction in overweight or obesity onset over 2-year follow-up relative to an alternative obesity prevention program and an obesity education control condition. Again, given that Project Health significantly reduced future onset of overweight/obesity in a prior trial and the present trial found that the body fat loss effects were significantly greater when Project Health is implemented in single-sex groups and paired with food response and attention training, there might be value in broadly implementing this combined intervention
Reviewer 3 Report
Enhancing Efficacy of a Brief Obesity and Eating Disorder Prevention Program: Long-Term Results from an Experimental 3 Therapeutics Trial by Eric Stice and co-workers
Interesting manuscript on the impact of the efficacy of an obesity/eating disorder prevention program, (Project Health) that incorporates food response inhibition and attention training (vs. sham) in single-sex and mixed sex groups in groups of young adults at risk of developing obesity. Groups administrated with food response and attention training showed significant reductions in body fat over a 2-year follow-up. However, this effect was more rapid and persistent in single-sex groups, whereas those who completed single- or mixed-sex Project Health groups plus sham training did not show a change in body fat.
In the introduction, the last paragraph anticipates the outcome of the study, and some material should be placed preferably in the discussion section.
In Table 1, the BMI of experimental groups does not seem homogeneous. Are there significant differences among study groups? Did the authors expect any effect because of these dissimilarities?
It is apparent that single-sex groups that received food response and attention training outperformed the rest of the experimental groups. There is no mention of gender differences. Are there differences between female-only groups vs. male-only groups in terms of response to the intervention?
Author Response
R3
Enhancing Efficacy of a Brief Obesity and Eating Disorder Prevention Program: Long-Term Results from an Experimental 3 Therapeutics Trial by Eric Stice and co-workers
Interesting manuscript on the impact of the efficacy of an obesity/eating disorder prevention program, (Project Health) that incorporates food response inhibition and attention training (vs. sham) in single-sex and mixed sex groups in groups of young adults at risk of developing obesity. Groups administrated with food response and attention training showed significant reductions in body fat over a 2-year follow-up. However, this effect was more rapid and persistent in single-sex groups, whereas those who completed single- or mixed-sex Project Health groups plus sham training did not show a change in body fat.
In the introduction, the last paragraph anticipates the outcome of the study, and some material should be placed preferably in the discussion section. In response, we edited the last paragraph of the introduction to make it clearer that we were describing the results reported in a previously published paper that described the acute effects from the present trial (p. 6).
In Table 1, the BMI of experimental groups does not seem homogeneous. Are there significant differences among study groups? Did the authors expect any effect because of these dissimilarities? It is true that there was some variation in body fat at baseline (we did not report BMI in Table 1). In response, we now note that the groups did not differ significantly on any study variables, including body fat (p. 11).
It is apparent that single-sex groups that received food response and attention training outperformed the rest of the experimental groups. There is no mention of gender differences. Are there differences between female-only groups vs. male-only groups in terms of response to the intervention? As requested, we tested whether the body fat change was significantly different for males versus females in single-sex Project Health groups paired with food response and attention training versus Project Health groups paired with sham training (p. 12). There was no evidence of moderation. Our original model fits slightly better than the moderation model. Additionally, none of the growth curves for response training differed by sex (i.e., all their p-values were > .05).